# Characterization of Macular Structural and Microvascular Changes in Thalamic Infarction Patients: A Swept-Source Optical Coherence Tomography–Angiography Study

**DOI:** 10.3390/brainsci12050518

**Published:** 2022-04-20

**Authors:** Chen Ye, William Robert Kwapong, Wendan Tao, Kun Lu, Ruosu Pan, Anmo Wang, Junfeng Liu, Ming Liu, Bo Wu

**Affiliations:** Department of Neurology, West China Hospital, Sichuan University, No. 37 Guoxue Lane, Chengdu 610041, China; yechen11@stu.scu.edu.cn (C.Y.); big_will_kwap@hotmail.com (W.R.K.); taowendan@wchscu.cn (W.T.); lkhxlc@163.com (K.L.); panruosu0601@163.com (R.P.); 2015181622040@stu.scu.edu.cn (A.W.); junfengliu225@outlook.com (J.L.); wyplmh@hotmail.com (M.L.)

**Keywords:** ischemic stroke, thalamic infarction, macula, microvasculature, SS-OCT/OCTA

## Abstract

Background: The retina and brain share similar neuronal and microvascular features. We aimed to investigate the retinal thickness and microvasculature in patients with thalamic infarcts compared with control participants. Material and methods: Swept-source optical coherence tomography (SS-OCT) was used to image the macular thickness (retinal nerve fiber layer, RNFL; ganglion cell-inner plexiform layer, GCIP), while OCT angiography was used to image the microvasculature (superficial vascular plexus, SVP; intermediate capillary plexus, ICP; deep capillary plexus, DCP). Inbuilt software was used to measure the macular thickness (µm) and microvascular density (%). Lesion volumes were quantitively assessed based on structural magnetic resonance images. Results: A total of 35 patients with unilateral thalamic infarction and 31 age–sex-matched controls were enrolled. Compared with control participants, thalamic infarction patients showed a significantly thinner thickness of RNFL (*p* < 0.01) and GCIP (*p* = 0.02), and a lower density of SVP (*p* = 0.001) and ICP (*p* = 0.022). In the group of patients, ipsilateral eyes showed significant reductions in SVP (*p* = 0.033), RNFL (*p* = 0.01) and GCIP (*p* = 0.043). When divided into three groups based on disease duration (<1 month, 1–6 months, and >6 months), no significant differences were found among these groups. After adjusting for confounders, SVP, ICP, DCP, RNFL, and GCIP were significantly correlated with lesion volume in patients. Conclusions: Thalamic infarction patients showed significant macular structure and microvasculature changes. Lesion size was significantly correlated with these alterations. These findings may be useful for further research into the clinical utility of retinal imaging in stroke patients, especially those with damage to the visual pathway.

## 1. Introduction

Strokes are one of the leading causes of mortality and long-term disability worldwide, and the economic cost of treatment and post-stroke care is substantial [1,2]. Accounting for 11% of posterior circulation infarcts [3] and 3–4% of cerebral ischemic events, thalamic infarction, both in isolation and combination with infarction lesions involving other structures, represents a frequent clinical entity [4,5].

As a part of the diencephalon and an important upstream and downstream fiber relaying center, the thalamus plays a vital role in managing sensory function, arousal, level of awareness, and certain cognitive functions [6,7,8]; importantly, the thalamus regulates the flow of visual, auditory and motor information. The thalamus receives blood supply from both anterior and posterior circulations of the brain, with several known variations [6]. Lesions in the thalamus have a variety of clinical syndromes depending on which vascular territories or thalamic nuclei are involved [9,10,11,12]. Numerous reports focusing on the variety of clinical syndromes and many comprehensive descriptions associated with thalamic stroke have been published [9,10,11,12]. Additionally, in addition to the typical sensory and cognitive impairments, thalamic infarction can also cause many vision-related symptoms, particularly due to the involvement of fibers or structures in the visual processing circuits [13,14,15,16]. Nonetheless, the underlying causes have been underexplored and the anatomical distribution of lesions caused by the various infarction mechanisms has been poorly defined.

Being an extension of the central nervous system (CNS), the retina shares histological, physiological, and embryological characteristics with the brain [17]. Currently, non-invasive measurements of the retinal neural tissue and microvasculature can be made in vivo, at a fast pace, and at a low cost by using optical coherence tomography (OCT) and OCT angiography (OCTA). Subtle neural and microvascular changes in the retina have been documented to reflect the pathological degeneration of the CNS during the disease process of an ischemic stroke [18,19,20] and other vascular origin diseases [21,22] through OCT/OCTA. Localized retinal nerve fiber layer defects (RNFLDs) were found to be associated with both new-onset and previous cerebral strokes [23]. Furthermore, significantly greater reductions in RNFL thickness were reported in cerebral posterior infarction patients [24]. By using magnetic resonance imaging (MRI) techniques, previous studies have focused on designated brain regions or lesion locations, and have found structural changes within the optic tracts after visual pathway insults (including optic tracts, thalamus, and occipital lobe) [25,26]. Particularly, a population-based study showed significant relationships between altered retinal thickness and changes in the visual pathway, especially the thalamus [27]. The main mechanisms were concentrated on trans-neuronal retrograde degeneration (TRD), hypoperfusion and hypoxia of small vessel networks. Based on the hypothesis of TRD, therapy of visual restoration training was explored in cortical blindness subjects with subtle therapeutic effects [28,29,30], but these studies were conducted on occipital stroke patients, and data of retinal changes were lacking. Additionally, potential bio- or imaging-markers are needed to assist better evaluation and guide treatments for visual deficit in stroke patients. The inner retina contains the neural integrity which forms direct synaptic connections between the thalamus and the microvasculature which reflect the cerebral microcirculations [17,31]. However, little is known about the specific macular structure and microvasculature changes in patients with thalamic infarction. Hence, our current study aimed to characterize the structural and micro-vascular changes in the macular of patients with thalamic infarction compared with age–sex-matched control participants using the swept-source OCT (SS-OCT)/SS-OCTA.

## 2. Materials and Methods

### 2.1. Study Participants and Clinical Data Collection

Ischemic stroke patients who were admitted to the Neurology Department of West China Hospital, China, were prospectively recruited. Patients were included in the study if they (1) had a clinical diagnosis of unilateral thalamic infarction confirmed by MRI (Figure 1); (2) could sit adequately and tolerate retinal imaging using the OCTA; (3) were classified as small-artery occlusion (SAA) stroke patients according to a modified TOAST classification [32]; and (4) provided written informed consent. The exclusion criteria were as follows: (1) diagnosed with diabetic retinopathy or other retinal diseases; (2) glaucoma; (3) a pacemaker or other contraindications for MRI examinations; (4) history of stroke or any other pathological conditions of CNS; or (5) poor MR or OCT/OCTA imaging quality. Age–sex-matched voluntary participants were recruited from the native communities in Chengdu as the control group. Control participants were included if they met the following criteria: (1) age 18 years or older; (2) can undergo and cooperate with retinal and MR imaging; (3) no history of cerebrovascular diseases, neurodegenerative diseases, or any other kind of central nervous system illness; and (4) no history of retinal diseases or ophthalmic abnormalities which could affect the retinal structure/microvasculature, such as severe glaucoma, severe cataract, age-macular degeneration, optic neuritis, or myopia.

Written informed consent was obtained from each participant or their legal guardians and approval of our project was obtained from the Ethics Committee of West China Hospital of Sichuan University [No. 2020 (922)].

Demographic and clinical information was collected in a standardized format, including gender, age, risk factors of cerebrovascular disease (history of hypertension, diabetes, and dyslipidemia), and current treatments (antiplatelets, anticoagulants, antihypertension, lipid-lowering, and antidiabetic drugs). Standard neurological examinations including eye movement and visual field were conducted by experienced neurologists or senior neurology residents. National Institute of Health Stroke Scale (NIHSS) scores were also documented.

### 2.2. MRI Acquisition and Procession

All patients underwent brain MRI scanning using a 3.0 T MR scanner (SIGNA^TM^ Premier, GE Medical Systems) with a 48-channel head coil. Head motion and scanner noise was reduced by using comfortable foam padding and earplugs. Structural MR imaging of high-resolution T1-weighted images was acquired using a brain volume (BRAVO) sequence with parameters as follows: repetition time (TR)/echo time (TE) = 7.2/3.0 ms; field of view = 256 × 256 mm; matrix = 256 × 256; slice thickness = 1.0 mm, no gap; flip angle = 12°; 152 slices. A diffusion-weighted image (DWI), T2-weighted image, fluid-attenuated inversion recovery (FLAIR) image, and non-contrast-enhanced 3D time of flight magnetic resonance angiography (3D TOF MRA) image were also acquired. The infarction lesion masks were drawn manually, based on the structural MR images (high-resolution 3D-T1) combined with DWI and FLAIR images by trained researchers using MRIcron [33]. Then, lesion volumes were obtained by the volume of interests (VOI). Experienced neurologists were consulted when disagreement occurred.

### 2.3. SS-OCT/OCTA Examination and Analysis

SS-OCT (VG 200; SVision Imaging Limited, Luoyang, China) was used to scan and image the macula of all participants. The imaging tool used a scan rate of 200,000 A-scans per second, and a central wavelength of 1050 nm (full width of 990–1100 nm). The OCT(A) tool had an axial resolution of 5 μm in the macula tissue. Using a three-dimensional protocol, high-resolution images (384 × 384 B-scans) of the macula (3 × 3 mm) were imaged; the in-built eye-tracking software was used to reduce projection artifacts while preserving the true layout [34,35]. An inbuilt algorithm in the SS-OCT was used to segment and measure the thickness of the retinal nerve fiber layer (RNFL) and ganglion cell–inner plexiform (GCIP) thicknesses (Figure 2A). The average RNFL and GCIP thicknesses measured in micrometers (µm) were used for our data analysis.

SS-OCT equipped with angiography (SS-OCTA) was also used to image and segment the macula into its three macular plexuses: the superficial vascular plexus, SVP; intermediate capillary plexus, ICP; and deep capillary plexus, DCP (Figure 2B).

The SVP was defined as the microvasculature between the base of the retinal nerve fiber layer (RNFL) to the junction between the inner plexiform layer (IPL) and inner nuclear layer (INL) as shown in Figure 2; ICP was defined as the microvasculature between the IPL/INL junction to the junction between INL and outer plexiform layer (OPL); DCP was defined as the microvasculature between the INL/OPL junction to 25 µm below it (Figure 2).

An examiner observed the segmentation of each image. The quality of the macular images was assessed objectively and subjectively, rejecting images with a signal quality less than 7 on a scale that goes up 10 [36]. *En face* angiograms with artifacts, blurry images, and images that revealed the presence of retinal diseases such as age-related macular degeneration (AMD) and macula edema were also excluded.

Microvascular density, which is defined as the percentage area (%) that was occupied by the microvasculature in the annulus region of measurement (3 × 3 mm around the fovea), was used to assess the microvasculature of the three macula plexuses. Measurement of the microvascular density was made by an inbuilt software in the OCTA tool.

The data acquisition and report from the SS-OCT(A) followed the Advised Protocol for OCT Study Terminology and Elements (APOSTEL) recommendations [37]. All the OCT/OCTA examinations and analyses were performed by an experienced researcher with a neuro-ophthalmology background.

### 2.4. Statistical Analysis

Continuous variables with normal distribution were expressed as mean ± standard deviation (SD), while skewed distribution was expressed medians and interquartile ranges. Categorical variables are presented as frequencies and percentages. Participants’ demographic variables were assessed using a chi-square test for categorical variables and an unpaired *t*-test or analysis of variance (ANOVA) test for continuous variables. Differences in SS-OCT/SS-OCTA parameters between each group were assessed using a generalized estimating equation (GEE) while adjusting for hypertension, diabetes, dyslipidemia, age, and gender. Correlations between these parameters and radiological and clinical features were also found with multivariable linear regression while adjusting for risk factors. All data were analyzed with SPSS (version 23; SPSS, Inc., Chicago, IL, USA), and *p* < 0.05 was considered statistically significant.

## 3. Results

### 3.1. Baseline Characteristics

From September 2020 to October 2021, a total of 71 participants (40 thalamic infarction patients and 31 controls) were enrolled in our study; however, five thalamic infarction patients were excluded due to poor MR image quality or diagnosis of another kind of pathological CNS disease. Our final data analysis included 35 thalamic infarction patients (24 males, mean age = 60.26 ± 9.37 years) and 31 age-sex matched control participants (20 males, mean age = 60.03 ± 6.71 years). The demographic and clinical characteristics are shown in Table 1. All our thalamic infarction patients were classified as having experienced a SAA ischemic stroke without evidence of large artery stenosis-occlusion or cardioembolism; 21 had a history of hypertension, 12 had type 2 diabetes mellitus, and 10 had dyslipidemia. All the thalamic infarction patients received antiplatelet and lipid-lowering treatments, 21 received antihypertension treatments, and 12 received treatments of antidiabetic drugs (9 for metformin, 2 for acarbose, 2 for dapagliflozin, 1 for linagliptin, 1 for miglitol, 1 for glyburide, and 4 for insulin). Of the control participants, 14 had a history of hypertension, 2 had type 2 diabetes mellitus, and 6 had dyslipidemia. One of them received antiplatelets, 14 received antihypertension treatments, 6 received lipid-lowering drugs, and 2 received antidiabetic drugs (all for metformin; no sulfonylurea, insulin, or any other kind of hypoglycemic agents were administered). The median NIHSS score of thalamic infarction patients was 1 (IQR = 0–4) point and lesion volume was 0.32 cm^3^ (IQR = 0.11–0.58).

### 3.2. SS-OCT/SS-OCTA Changes among the Groups

Thalamic infarction patients showed thinner RNFL (17.388 ± 1.759 vs. 19.286 ± 1.162, *p* < 0.001; Table 2) and GCIP (63.577 ± 8.298 vs. 71.277 ± 6.156, *p* = 0.006; Table 2) thicknesses compared with the control group. As shown in Table 2, compared with control participants, thalamic infarction patients showed significantly reduced SVP (0.202 ± 0.025 vs. 0.219 ± 0.019, *p* = 0.001) and ICP (0.169 ± 0.018 vs. 0.186 ± 0.015, *p* = 0.022) densities. No significant difference (*p* = 0.763, Table 2) was seen in the DCP when the two groups were compared.

Thalamic infarction patients were stratified according to the location of infarction (i.e., left or right cerebral hemisphere); the hemisphere with infarction was described as the ipsilateral side while the hemisphere without infarction was described as the contralateral side. When compared with eyes on the contralateral side, ipsilateral eyes had reduced density of SVP (0.198 ± 0.023 vs. 0.205 ± 0.026, *p* = 0.033; Table 3), thinner RNFL (17.194 ± 1.742 vs. 17.552 ± 1.784, *p* = 0.010; Table 3) and GCIP (62.397 ± 8.102 vs. 64.579 ± 8.454, *p* = 0.043; Table 3). There was no significant difference (*p* > 0.05, Table 3) in ICP and DCP between the two groups.

Thalamic infarction patients were also sub-grouped according to the duration of disease (Group 1: <1 month, Group 2: 1–6 months, Group 3: >6 months) as shown in Appendix A and Figure 3. Group 1 patients showed significantly sparser SVP when compared with Group 2 (0.198 ± 0.022 vs. 0.203 ± 0.022, *p* = 0.013) and Group 3 (0.198 ± 0.022 vs. 0.198 ± 0.035, *p* = 0.042). Group 2 showed a denser ICP when compared with Group 1 (0.167 ± 0.016 vs. 0.163 ± 0.026, *p* = 0.004) and a sparser ICP when compared with Group 3 (0.016 vs. 0.163 vs. 0.175 ± 0.016, *p* = 0.004).

### 3.3. Correlations of SS-OCT/OCTA Parameters with Lesion Size and Disease Duration in Thalamic Infarction Group

Correlations between SS-OCT/OCTA parameters and radiological and clinical features (lesion size and disease duration) were further explored in thalamic infarction patients. After adjusting for confounders (age, gender, and vascular risk factors, i.e., hypertension, diabetes, and dyslipidemia)), SS-OCT/SS-OCTA parameters significantly correlated with lesion volume (all *p* < 0.05; Table 4), while there was no significant correlation found with disease duration (*p* > 0.05; Table 4).

## 4. Discussion

In the present study, we found that thalamic infarction patients showed sparser macular microvasculature and thinner macular thicknesses when compared with our control group. The results also showed that eyes on the ipsilateral side displayed sparser SVP and thinner RNFL and GCIP when compared with contralateral eyes. Importantly, thalamic infarction lesion size significantly correlated with macular microvasculature and structural changes.

RNFL and GCIP as assessed by OCT macular scanning reflect the in vivo condition of the axons, dendrites and cell bodies of the retina [38]; RNFL and GCIP, which form the retinal ganglion cell (RGC), play a vital role in visual processing in the retina [39]. Interestingly, in our study, we found the RNFL and GCIP to be significantly thinner in thalamic infarct patients when compared with stroke-free control participants; thus, thalamic infarction may lead to retinal neuro-axonal damage. Damage to the thalamus, the principal region involved in visual processing, may result in damage to connections within the visual tract, thereby causing regressive neuro-axonal damage of the optic nerve, ultimately ending in the thinning of the RGCs because of its connection to the optic nerve. Undeniably, patients who have experienced a thalamic stroke frequently experience visual complaints [6,18,40], implicating that structural changes in the retinal thickness may contribute to the neuro-axonal damage along the visual pathways. Generally, we suggest that thalamic abnormalities may be reflected in the retina as thinner RNFL and GCIP because of its relationship with the thalamus. Contrarily, it may be plausible that ganglion cell death may cause anterograde degeneration, resulting in RNFL and GCIP thinning and ultimately resulting in changes in the thalamus [41]. However, further studies are needed to validate our hypothesis.

The pathophysiological process underlying the significantly altered macular microvasculature in thalamic infarction patients compared with control patients is uncertain. Neuroimaging reports showed that thalamic stroke patients have microvascular emboli [40,42] and increased cerebral small vessel disease [4]. Additionally, atherosclerosis, a pathophysiological cause of thalamic stroke, has been reported to cause neurodegeneration, which may lead to decreased retinal microvascular densities and reduced retinal fractals compared with healthy controls [4]. Therefore, we speculate that sparser macular microvasculature and reduced microvascular densities in thalamic stroke patients may reflect neurodegeneration with associated microvascular impairment. Importantly, our results showed that microvasculature impairment was more sensitive in the SVP (superficial microvasculature) than the ICP and DCP (deeper microvasculature). Retinal microvascular reflects damage of cerebral microcirculation [43] and during the ischemic injury, the superficial plexus has been suggested to be more severely impaired than the deeper plexuses because it is the entry point of blood flow into the retina [44,45]; of note, it has been suggested that the superficial retinal vasculature is a risk indicator of ischemic stroke [46] and is associated with the incidence of lacunar stroke [47]. This may explain why microvascular impairment was more sensitive in the SVP than in the ICP and DCP in our study. Additionally, atherosclerosis, the key factor of thalamic infarction, has been detected in blood vessels and more prominently in the retinal arteries [48]. Since SVP reflects the arterial circulation of the retina [49], while the deeper microvascular plexuses reflect the venular circulation [50], this may also explain why SVP impairment was more sensitive than the deeper plexuses.

Eyes on the ipsilateral side of thalamic infarction showed sparser SVP and thinner macular thickness compared with eyes on the contralateral side. Regarding the pathophysiology of our findings, the location of cerebral infarction may lead to severe retinal microvascular impairment and neurodegeneration. For example, damage to the brain may result in damage to the retinal structure and microvasculature, as shown in previous reports [51,52,53]. Altogether, findings from previous studies suggest that certain abnormalities in the brain may be reflected in the retina as microvascular impairment and neurodegeneration.

Patients with thalamic infarction of less than a month showed significantly sparser SVP when compared with patients with infarction of more than a month. Contrarily, patients with thalamic infarction for more than a month but less than 6 months showed denser ICP when compared with patients with a duration of less than a month and showed a sparser ICP when compared with patients with a duration of more than 6 months. Infarction leads to deprivation of oxygen in the brain; thus, duration is critical in the management of cerebral infarction [54]. Our report suggests that microvascular changes occur over different durations; nonetheless, we did not observe a significant correlation between the SS-OCT/SS-OCTA parameters and disease duration. Future studies with larger sample sizes of homogenous thalamic infarction patients are needed to validate our hypothesis.

Interestingly, we showed that microvascular impairment (sparser microvasculature) and neurodegeneration (thinner macular structure) significantly correlated with thalamic lesion volume. Clinically, infarct lesion volume reflects neurological damage. Since thalamic infarction leads to microvascular and neurological damage, the correlation between lesion volume and retinal microvascular impairment and neurodegeneration may indicate that the retina may have a role as a screening biomarker of thalamic infarction.

Currently, although some visual restoration training and compensatory therapy strategies were conducted in patients with issues with their visual circuits [55,56], individual heterogeneity and variability exist in the efficacy of such treatments [28]. It is important to accurately identify groups who may benefit from these treatments. In addition, apart from the routine secondary prevention treatments of cerebrovascular disease, there is still a lack of individualized treatment and evaluation indicators for cerebral infarction patients with different clinical syndromes. The present study illustrates some significant alterations of retina structure and microvasculature after thalamic infarction and explored their correlations with damage to the brain and time effects. These findings highlight the potential of SS-OCT/OCTA indices as markers for disease assessment and response to the therapy in these patients, and suggest the need for research on related interventions. However, this study has several limitations. Firstly, a major limitation of our current study is the small sample size and inclusion of only Chinese participants from a single center. Second, no symptoms of the visual field or oculomotor deficits occurred in these patients, which reduced the clinical interest and significance to some extent; however, as a type of ischemic stroke with a lower proportion, only 11.7% of thalamic infarction patients developed neuro-ophthalmologic manifestations in a long-time cohort [5]. Further studies with a larger sample size are needed. Thirdly, previous studies have shown that the degree of TRD was time dependent [23,24,25,57]. In our study, there were no significant changes among groups who underwent different durations of treatment. The lack of a comparison at different time-points for the same person and the small number of participants may be the reasons for this negative result. Long-term follow-up cohort studies are needed in the future.

## 5. Conclusions

In conclusion, our study showed that patients with thalamic infarction have significantly thinner sub-retinal layers, and impaired macular microvasculature compared with controls. We also showed that an altered macular structure and microvasculature significantly correlated with infarction lesion volumes. The findings of our study emphasize the importance of further research into retinal imaging as a potential indicator for thalamic infarction. Longitudinal studies with a greater sample size are needed to validate our hypotheses.

## Figures and Tables

**Figure 1 brainsci-12-00518-f001:**
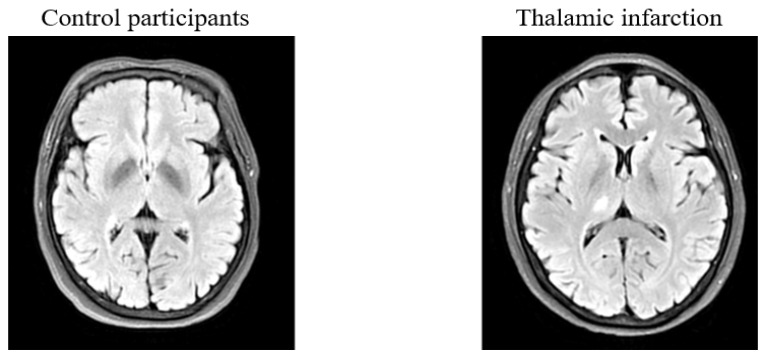
Representative MR imaging of thalamic infarction patient and control participant. The **right** image shows a fluid-attenuated inversion recovery (FLAIR) MR image of a patient with an ischemic stroke lesion beside the right thalamus, while the **left** image shows a FLAIR image of control.

**Figure 2 brainsci-12-00518-f002:**
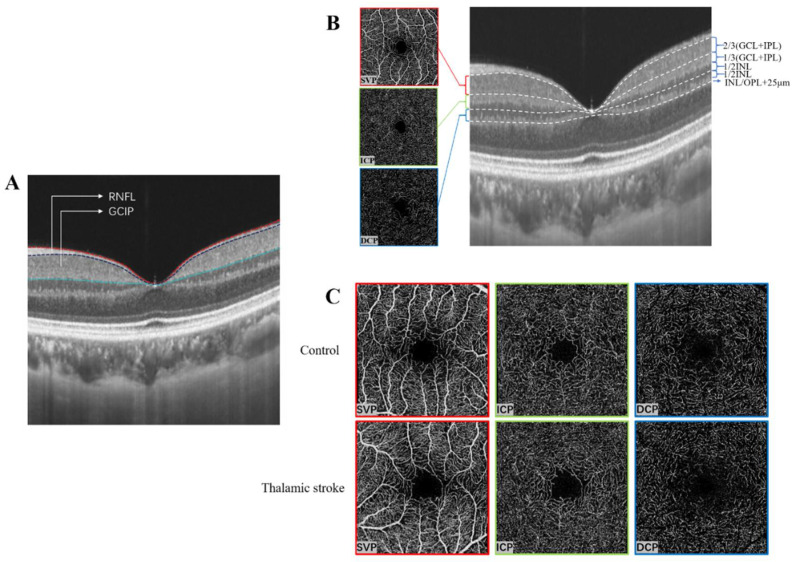
Segmentation of macular structure and microvasculature. (**A**) shows the macular structural segmentation. (**B**) shows the segmentation of the macular microvasculature. The SVP was defined as the microvasculature between the base of the retinal nerve fiber layer (RNFL) to the junction between the inner plexiform layer (IPL) and inner nuclear layer (INL). ICP was defined as the microvasculature between IPL/INL junction to the junction between INL and outer plexiform layer (OPL). DCP was defined as the microvasculature between the INL/OPL junction to 25 µm below the OPL. (**C**) shows the *en face* OCTA images between thalamic stroke patients and healthy controls.

**Figure 3 brainsci-12-00518-f003:**
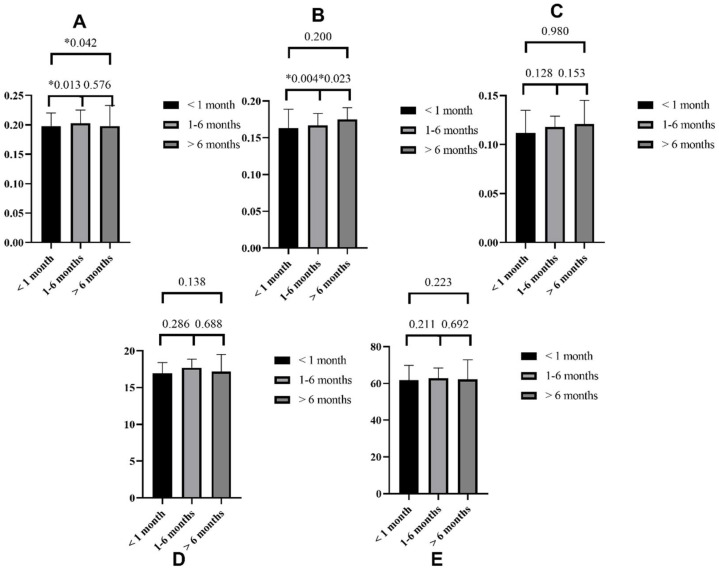
Differences in SS-OCT/OCTA parameters among three groups based on the disease duration (Group 1: <1 month, Group 2: 1–6 months, and Group 3: >6 months). (**A**) for SVP, (**B**) for ICP, (**C**) for DCP, (**D**) for RNFL, and (**E**) for GCIPL. * ***p*** < 0.05.

**Table 1 brainsci-12-00518-t001:** Demographics and clinical information.

	Thalamic Infarction (*n* = 35)	Control Participants (*n* = 31)
Age, years	60.26 ± 9.37	60.03 ± 6.71
Gender (male), n	24/11	20/11
SBP, mmHg	135.40 ± 22.51	134.96 ± 3.94
DBP, mmHg	84.11 ± 16.03	72.90 ± 8.46
Hypertension, n	21	14
Type 2 Diabetes, n	12	2
Dyslipidemia, n	10	6
Current treatments, n		
Antiplatelets	35	1
Anticoagulants	0	0
Antihypertension	21	14
Lipid-lowering	35	6
Antidiabetic drugs	12	2
Duration, months	0.5 (0.1–8.5)	-
NIHSS score	1 (0–4)	-
Lesion volume, cm^3^	0.32 (0.11–0.58)	-

Data are presented as means ± standard deviation or median with 25–75% quartiles. SBP, systolic blood pressure; DBP, diastolic blood pressure; NIHSS, National Institute of Health Stroke Scale scores.

**Table 2 brainsci-12-00518-t002:** Comparison of SS-OCT/SS-OCTA parameters between thalamic infarction and control group.

	Thalamic Infarction	Control Group	*p*
SVP, mm^2^	0.202 ± 0.025	0.219 ± 0.019	**0.001**
ICP, mm^2^	0.169 ± 0.018	0.186 ± 0.015	**0.022**
DCP, mm^2^	0.118 ± 0.020	0.123 ± 0.013	0.763
RNFL, µm	17.388 ± 1.759	19.286 ± 1.162	**<0.001**
GCIPL, µm	63.577 ± 8.298	71.277 ± 6.156	**0.002**

SVP, the superficial vascular plexus; ICP, intermediate capillary plexus; DCP, deep capillary plexus; RNFL, retinal nerve fiber layer; GCIPL, ganglion cell–inner plexiform. Data adjusted for age, gender, and vascular risk factors (hypertension, diabetes, and dyslipidemia). Values in bold indicate ***p*** < 0.05.

**Table 3 brainsci-12-00518-t003:** Comparison of SS-OCT/SS-OCTA parameters between ipsilateral and contralateral eyes.

	Ipsilateral Eyes	Contralateral Eyes	*p*
SVP, mm^2^	0.198 ± 0.023	0.205 ± 0.026	**0.033**
ICP, mm^2^	0.169 ± 0.015	0.169 ± 0.021	0.650
DCP, mm^2^	0.119 ± 0.020	0.117 ± 0.020	0.238
RNFL, µm	17.194 ± 1.742	17.552 ± 1.784	**0.010**
GCIPL, µm	62.397 ± 8.102	64.579 ± 8.454	**0.043**

SVP, the superficial vascular plexus; ICP, intermediate capillary plexus; DCP, deep capillary plexus; RNFL, retinal nerve fiber layer; GCIPL, ganglion cell–inner plexiform. Data adjusted for age, gender, and vascular risk factors (hypertension, diabetes, and dyslipidemia). Values in bold indicate ***p*** < 0.05.

**Table 4 brainsci-12-00518-t004:** Correlation of lesion volume and duration with SS-OCT/SS-OCTA parameters in patients.

	Lesion Volume		Duration, Months	
	B (95% CI)	*p*	B (95% CI)	*p*
SVP, mm^2^	0.019 (0.010–0.028)	**<0.001**	−0.001 (−0.003–0.001)	0.321
ICP, mm^2^	0.004 (0.001–0.008)	**0.026**	0.0004 (−0.0005–0.001)	0.372
DCP, mm^2^	−0.011 (−0.015–−0.007)	**<0.001**	0.0003 (−0.001–0.0003)	0.278
RNFL, µm	1.321 (0.422–2.219)	**0.004**	−0.022 (−0.123–0.08)	0.678
GCIPL, µm	7.226 (2.641–11.811)	**0.002**	−0.285 (−0.877–0.307)	0.346

SVP, the superficial vascular plexus; ICP, intermediate capillary plexus; DCP, deep capillary plexus; RNFL, retinal nerve fiber layer; GCIPL, ganglion cell–inner plexiform. Data adjusted for age, gender, and vascular risk factors (hypertension, diabetes, and dyslipidemia). Values in bold indicate ***p*** < 0.05.

## Data Availability

The data that support the findings of this study are available from the corresponding author upon reasonable request.

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
