# Peer review of "Characterization of Macular Structural and Microvascular Changes in Thalamic Infarction Patients: A Swept-Source Optical Coherence Tomography–Angiography Study"

_brainsci, 2022, doi:10.3390/brainsci12050518_

Round 1
Reviewer 1 Report
This is a well prepared and executed study.
Introduction should be more focused on retinal changes in cerebrovascular diseases than on general epidemiology of stroke and techical nuaces of OCT or use of MRI in stroke.
There are numerous formal erros in sentences i.e. 83 page 2 "Age-sex matcher". Please correct.
How did the authors recruited the control group ? Please explain
There is a substantial disproprotion of diabetic patients in control and study group- this is a bias that might strongly influence the differences in the results of OCT
More clinical details should be shown for compaired patients: i.e. type of diabetes, duration of the diseases, drugs thath patients were recieving for thoese diseases (metformin, sulfonylourea, insulin ?), anticoagluation status etc.
In figure 3 i would rather see larger figures and Group 1-3 should be repleced by the actual name of the group.
Table 4 doese not make any sence- there is no acctual figure for the months or volume- only B and p values. Please correct
Author Response
Response to Reviewer 1 Comments:
Point 1: Introduction should be more focused on retinal changes in cerebrovascular diseases than on general epidemiology of stroke and techical nuaces of OCT or use of MRI in stroke.
Response 1: We thank the reviewer for the very helpful and constructive comment. Indeed, it is more in accordance with the content of our manuscript for focusing on retinal changes in cerebrovascular diseases in the Instruction section. Regarding this suggestion, we have shortened the descriptions about OCT or MRI use in stroke and then added contents that are concentrated on previous reports about retina changes after stroke (see Line 52-61, page 2 in revised version).
Point 2: There are numerous formal erros in sentences i.e. 83 page 2 "Age-sex matcher". Please correct.
Response 2: We are sorry for not being cautious enough in the early version. In the revised version, we have carefully reviewed the article and made corrections on such errors.
Point 3: How did the authors recruited the control group? Please explain
Response 3: We recruited voluntary elderly persons as the control group in the local communities through online platforms and posters. In the revised version, we have added descriptions about the recruitment in the Materials and Methods- Study participants and clinical data collection section.
Point 4: There is a substantial disproprotion of diabetic patients in control and study group- this is a bias that might strongly influence the differences in the results of OCT.
Response 4: We are thankful for the reviewer’s critic comment very much. In fact, as we described above, we recruited the stroke-free elderly persons as control group from the local communities in our city. Diabetes, hypertension, hyperlipidemia and other chronic vascular risk conditions are common among the elderly in China nowadays. Meanwhile, elderly controls recruited in the community in this way may better reflect real-world conditions. Given this, we did not define them in the manuscript as healthy controls, but as control participants (Mutlu U, et al. Hum Brain Mapp. 2018; Mutlu U, et al. Neurobiol Aging. 2017.). As for the possible bias of these conditions, firstly, we excluded subjects with retinal diseases or ophthalmic abnormalities which could affect the retinal structure/microvasculature by an experienced expert with a neuro-ophthalmology background through OCT examination at the time of inclusion. In addition, we adjusted for confounding factors that may affect the fundus, including diabetes, in the subsequent analyses. Please see the details in Materials and Methods-Study participants and clinical data collection, SS-OCT /OCTA examination and analysis, and Statistical Analysis section in the revised version.
Point 5: More clinical details should be shown for compaired patients: i.e. type of diabetes, duration of the diseases, drugs thath patients were recieving for thoese diseases (metformin, sulfonylourea, insulin?), anticoagluation status etc.
Response 5: Thanks again for the reviewer’s constructive advice. More clinical details are added in the Results section, Line 193-202, page 5 and Table 1, page 5-6 in the revised version.
Point 6: In figure 3 i would rather see larger figures and Group 1-3 should be repleced by the actual name of the group.
Response 6: Thanks for the reviewer’s constructive advice. In the revised version, we replaced a new larger Figure 3 with actual name of each group, please see the page 7.
Point 7: Table 4 doese not make any sence- there is no acctual figure for the months or volume- only B and p values. Please correct
Response 7: We are grateful for the reviewer’s comment. As a matter of fact, we further explored the correlations between retina OCT/OCTA parameters and disease duration and lesion volume of patients with multivariable linear regression, while adjusting for confounding factors. In order to illustrate these results more clearly, we amend the descriptions in Line 248-252, page 7 and added 95% credible interval (CI) in the Table 4. The actual values of disease duration(months) and lesion volume(cm3) are presented as parts of baseline characteristics in Table 1, page 6.
Reviewer 2 Report
Very important area of research. This will definitely benefit the future research and treatment.
Author Response
We deeply appreciate the kind encouragement from the reviewer. And we will conduct more in-depth research on mechanisms and interventions in the future.
Reviewer 3 Report
The authors aimed to compare to investigate the retinal thickness and microvasculature in patients with thalamic infarcts compared with control participant. The idea to investigate the visual disease in stroke patients is important, at the light of the lack of literature about. The paper is well written, the introduction is clear and explain the aim of the study, the methods are adequate to study design and the discussion is coherent with the results. At the best of my knowledge, there are some minor issue to address:
Introduction:
- It could be interesting to explain if in literature there are information about therapeutic option of visual deficit in stroke patients (eg. surgical option, drugs therapy, visual rehabilitation)
Material and Methods:
- The authors talk about stroke but not specify the type (emorragic or ischemic).
- Please, add in the table 1 the gender (male and female)
Results:
- None
Discussion:
- it would be desirable to define the clinical implication and possible therapeutic option based on your result in order to stimulate a new line of research in field
Best Regards
Author Response
Response to Reviewer 3 Comments:
Point 1: introduction: It could be interesting to explain if in literature there are information about therapeutic option of visual deficit in stroke patients (eg. surgical option, drugs therapy, visual rehabilitation).
Response 1: We thank the reviewer for the very helpful and constructive comment. Indeed, it is important to explain some information about therapeutic option of visual deficit in stroke patients in the Instruction section. Regarding this suggestion, we amended introduction part and added some description about mechanisms and interventions in the relevant area. Please see Line 61-75, page 2 in the revised version.
Point 2: Material and Methods: The authors talk about stroke but not specify the type (emorragic or ischemic). Please, add in the table 1 the gender (male and female).
Response 2: Thanks for the reviewer’s constructive advice. It’s our mistake for not clarifying this issue. In the present study, we included patients with thalamic infarctions, so all stroke type of the patients should be ischemic. We amended the description and added the term “ischemic” in this section, please see Line 82, page 2.
Point 3: Discussion: it would be desirable to define the clinical implication and possible therapeutic option based on your result in order to stimulate a new line of research in field
Response 3: We are grateful for the reviewer’s valuable comment very much. We modified the discussion part and added description about clinical implication and possible therapeutic option based on our results, please see Line 329-351, page 9 in the revised version.